# Risk perceptions and preventive practices of COVID-19 among healthcare professionals in public hospitals in Addis Ababa, Ethiopia

**Wakgari Deressa**[1]*, **Alemayehu Worku**[1], **Workeabeba Abebe**[2], **Muluken Gizaw**[1], **Wondwossen Amogne**[3]

**1** Department of Preventive Medicine, School of Public Health, College of Health Sciences, Addis Ababa University, Addis Ababa, Ethiopia, **2** Department of Pediatrics and Child Health, School of Medicine, College of Health Sciences, Addis Ababa University, Addis Ababa, Ethiopia, **3** Department of Internal Medicine, School of Medicine, College of Health Sciences, Addis Ababa University, Addis Ababa, Ethiopia

* deressaw@gmail.com

## Abstract

Healthcare professionals are at higher risk of contracting the new coronavirus disease 2019 (COVID-19). Although appropriate preventive measures are the most important interventions to prevent coronavirus infection among healthcare workers, they are also highly concerned about the consequences of the pandemic. The aim of this cross-sectional study was to assess preventive practices, perceived risk and worry about COVID-19 crisis among healthcare professionals at six public hospitals in Addis Ababa, Ethiopia. A systematic random sampling technique was used to select 1,134 respondents (52.6% females). Data were collected between 9th and 20th June 2020 using self-administered questionnaires. Descriptive statistics were used to summarize the data. A multiple linear regression analysis was performed to identify factors associated with worry about COVID-19 crisis. The highest percentage of respondents were nurses (39.3%) and physicians (22.2%), followed by interns (10.8%) and midwives (10.3%). Wearing facemask (93%) and frequent hand washing (93%) were the commonly reported preventive practices. Perceived risk of becoming infected with coronavirus (88%) and the potential risk of infection to their family (91%) were very high. The mean (median) worry score about COVID-19 crisis was 2.37 (3.0), on 1 to 3 scale, with 1 (not worried) and 3 (highly worried). The majority worried a lot about the health system being overwhelmed by COVID-19 patients (92%), the health of their loved ones (90%) and losing someone due to COVID-19 (89%). Respondents who had previously provided clinical care to Ebola, SARS and cholera patients had significantly lower levels of worry about COVID-19 crisis than participants who had not (β = -1.38, P<0.001). Our findings reveal respondents' widespread practice of preventive measures, highest levels of perceived risk and worry about the COVID-19 crisis. Increased perceived risk and worry about COVID-19 might enable healthcare workers to adopt appropriate preventive measures more effectively against the disease.

**Data Availability Statement:** All relevant data are within the manuscript and its Supporting Information files.

**Funding:** This study was funded by Addis Ababa University.

**Competing interests:** The authors declare no competing interests.

# Introduction

The novel coronavirus disease 2019 (COVID-19) that was declared as a pandemic by the World Health Organization (WHO) on the 11[th] of March 2020 [1] has affected over 123 million people and has caused more than 2.7 million deaths globally as of 21[st] March 2021 [2]. This new severe acute respiratory syndrome coronavirus 2 (SARS-CoV-2) has rapidly spread to all countries and territories around the world. Up to 7[th] March 2021, Ethiopia reported a total of 166,138 confirmed COVID-19 cases and 138,500 recoveries from over 2,178,403 total tests, among whom 2,429 have died [3]. Over 1,311 health workers have contracted coronavirus in Ethiopia as of 17[th] September 2020.

Healthcare professionals (HCPs) are highly vulnerable to SARS-COV-2 infection due to their clinical role in the healthcare settings [4]. Most HCPs are working in isolation units, critical care units, intensive care units (ICUs), emergency units, working in frontline positions, and having contact with suspected and confirmed COVID-19 cases. During the early stage of COVID-19 pandemic in the USA, the prevalence of SARS-CoV-2 infection among healthcare workers was 7.3% and particularly, infections were most common among nurses [5]. In The Netherlands, 96 (5%) of 1,796 healthcare workers screened in three hospitals were tested positive for SARS-CoV-2 just 10 days after the first reported COVID-19 case in the country [6].

During the early stage of the pandemic, more than 278 physicians from almost all medical specialties have died due to COVID-19 with the majority (44%) from Italy mainly because of lack understanding of the transmission mechanisms of the virus and its preventive measures [7]. Studies in China reported 3,387 COVID-19 cases among healthcare workers (4.4% of all cases), with 23 attributable deaths [8]. In some countries such as Spain, they have reported that 13% to 14% of the country's cases were in HCPs [9]. Overall, as much as 10% of HCPs are infected with SARS-CoV-2 in many countries [4] and the WHO has developed infection prevention and control guidance to be implemented at the national and healthcare facility level in order to reduce coronavirus infection among HCPs [10].

Studies have identified major sources of worry and anxiety among HCPs particularly due to lack of appropriate personal protective equipment (PPE); being exposed to COVID-19 at work; not having rapid access to testing if they develop COVID-19 symptoms, and fear of spreading infection at work; uncertainty that their organization will support/take care of their personal and family needs if they develop infection; access to childcare during increased work hours and school closures; and support for other personal and family needs as work hours and demands increase [11].

A recent study from China reported the challenges facing frontline healthcare workers during the COVID-19 pandemic which includes high risk of infection, insufficient PPE, heavy workloads and manpower shortages, confusion, discrimination, isolation, separation from families, and burnout [12]. Studies also show that many HCPs are highly worried about being infected by coronavirus, and are most concerned about spreading the virus to their family and loved ones, or to the vulnerable clients in the hospital or the community [11]. Under these stressful conditions, HCPs have been challenged to be effectively engaged in the fight against COVID-19. Furthermore, studies conducted during the early stages of the pandemic have suggested that perceived personal risk of infection, worry about COVID-19 and the detrimental health effects are linked to improved protective behaviors [13].

Since the first case of COVID-19 was reported in Ethiopia on 13 March 2020, the Ministry of Health in collaboration with its partners, conducted different trainings on preventive measures for HCPs at several hospitals and health centers, with supplies of PPE materials. The HCPs in Ethiopia have worked tirelessly and played a crucial role in the management of COVID-19 cases, despite high personal risks and worries about the current pandemic crisis. However, no study has been undertaken in the country on risk perception and preventive

practices of HCPs during the current COVID-19 pandemic. In addition, emotional reactions and feelings of healthcare workers such as worries about COVID-19 crisis have not been studied. To address this research gap, this study was conducted to assess preventive practices, perceived risk and worry about COVID-19 crisis among HCPs. Understanding the preventive practices, risk perception and worries of HCPs would help in protecting them and preventing the COVID-19 pandemic through effective risk communication.

## Methods and materials

### Study design and setting

This cross-sectional study was conducted from 9th to 26th June 2020 at six public hospitals in Addis Ababa, three months after the first confirmed COVID-19 case in Ethiopia. Addis Ababa city is the most populated urban city in the country, and had a population of about 3.6 million in 2019 [14]. The city also had better health infrastructure and the highest number of qualified medical personnel compared with any city or region in the country. There were 12 hospitals and close to 100 health centers belonging to the public sector, and about 25 private hospitals in Addis Ababa city. There were also over 17,000 HCPs of different categories in the city, including 2,441 (14%) physicians and 8,172 (47%) nurses. The hospitals selected for the current study provide outpatient and inpatient services for the city residents and patients coming from different parts of the country.

### Study population and sample size calculation

The study was conducted among HCPs working in the different clinical departments or units of six public hospitals in Addis Ababa, mainly obstetrics and gynecology, surgery, pediatrics, internal medicine, outpatient department (OPD), emergencies, ICUs, operation room/ward, screening/triage, laboratory and anesthesia. The six study hospitals included: Tikur Anbessa Specialized Hospital (TASH), Zewditu Memorial Hospital (ZMH), Ghandi Memorial Hospital (GMH), Menelik II Hospital (MH), Yekatit 12 Hospital Medical College (Y12HMC) and St. Paul Hospital Millennium Medical College (SPHMMC). The study population of this survey included intern and resident doctors, general practitioners, medical specialists and sub-specialists, health officers, anesthetists, nurses, midwives, pharmacists, laboratory technicians and technologists, physiotherapists and X-ray technicians, who directly or indirectly exposed to suspected or confirmed patients with COVID-19.

Since COVID-19 is a new disease, we assumed that at least 50% of study participants had higher risk perception regarding COVID-19, and the estimated sample size was calculated with 95% confidence limit, with 4% precision and a design effect of 1.5 using 20% non-response rate. Accordingly, the minimum total sample size calculated for this survey was 1,080 respondents, and finally, a total of 1,200 HCPs of all categories were targeted for the study.

### Sampling procedures

A systematic random sampling technique was applied to select the study participants. In the first stage, the six hospitals were purposively selected from 12 hospitals in the city. The total sample size was allocated to the selected hospitals proportional to their estimated HCPs at the time of the study. In the second stage, clinical departments or units were identified, and in the third stage, study participants were selected proportionally to the estimated number of HCPs listed in the different departments or units of the hospital, using systematic sampling. All eligible HCPs in each department/unit who consented to participate were selected into the study using systematic sampling.

## Survey instrument and data collection

A structured paper-based self-administered questionnaire was used to collect the data. The questionnaire was developed in English by the authors of the study by reviewing the previously conducted studies [15, 16] and adopting questions from the WHO website on prevention measures, risk perception and emotional reaction such as worry about COVID-19 crisis [17]. The questionnaire is composed of parts on the demographic (gender, age) and occupational characteristics of the respondents (hospital, department/unit, professional category, and work experience), preventive practices, clinical experience with COVID-19, Ebola and SARS, preparedness to combat COVID-19, risk perception, and worry about the current COVID-19 crisis. Questions related to measures taken to prevent infection from coronavirus included hand washing for at least 20 seconds, use of disinfectants, wearing facemask, staying home, physical distancing, covering mouth and nose while coughing and sneezing and other preventive measures. Most of the questions were dichotomous (yes/no).

Risk perception in this study was measured using three questions: concern about their personal health, perceived risk of being infected with coronavirus, and the potential risk to their family, loved ones or others due to their clinical role in the hospital. Responses for each question were rated on a five-point Likert scale, 1 = 'not worried at all', 2 = 'somewhat not worried', 3 = 'neutral', 4 = somewhat worried', and 5 = 'extremely worried'. The total score of the scale was the sum of the three items, ranging from 3 to 15. A higher total score indicates a greater perceived risk of COVID-19. However, the perceived risk index had low internal consistency (reliability) with Cronbach's alpha of 0.50. This low reliability of the perceived risk index of COVID-19 prevented from producing the total score for further analysis, which was only limited to the single item descriptive data.

The last part of the questionnaire comprised 12 questions related to how much respondents worried about the current COVID-19 crisis, such as losing someone they love due to the disease, health system being overwhelmed, their mental and physical health, etc., on a three-point scale, where 1 = 'don't worry at all', 2 = 'worry somehow', and 3 = 'worry a lot'. The total score of the scale ranged from 12 to 36. The internal consistency (reliability) of the questions was tested by applying Cronbach's alpha and the coefficient of the reliability of scale was estimated at 0.91, which is highly acceptable.

A total of 12 experienced data collectors with health backgrounds were involved in the data collection. A guideline was developed by the research team to guide the data collectors and supervisors for data collection, quality assurance of data and ethical conduct. Training and orientation on the survey tool and methodology including how to administer the questionnaire was conducted for the data collectors and supervisors using webinar on 2nd June 2020. After explaining the purpose of the study and obtaining written or oral informed consent, study participants were given a paper-based questionnaire at their workplace and the respondents filled out their own questionnaires.

The purpose of the study was clearly stated in the questionnaire and the study participants were asked to complete the questionnaire with honest answers. They were encouraged to fill out the questionnaire whilst the data collectors were still in the hospital during the data collection period. A collection center was also prepared in the Hospital Director's office to gather the questionnaires from the healthcare workers who were unable to directly deliver the completed questionnaires to the data collectors. The data collection took place simultaneously in the six hospitals. The questionnaires were finally checked for completeness and consistency upon collection. All responses were anonymous.

## Data analysis

Data were entered into the Census Surveys Professional (CSPro) Version 7.2 statistical software package and subsequently exported to SPSS version 23.0 (SPSS Inc., IBM, USA) for

cleaning and data analysis. Descriptive analysis was applied to calculate the frequencies, proportions and mean scores, and the results were presented as percentages for the categorical variables, and as the mean ± standard deviation (SD) for the quantitative variables.

Responses on a Likert scale of the healthcare workers' worries about the COVID-19 crisis were summed up (ranged from 12 to 36) and analyzed as a continuous value. A multiple linear regression analysis was carried out with the overall worry score about COVID-19 crisis as the outcome variable and the different individual and work-related characteristics as the potential predictors (gender, professional category, department/unit, hospital, preparedness to provide direct care to COVID-19 patients, ever provided care to COVID-19 patients, and previous exposure to Ebola, SARS or cholera outbreak/pandemic). Standardized beta-coefficients of the predictors in the simultaneous regression model and the 95% confidence intervals were estimated to investigate the association between the outcome variable and the predictors. An alpha level of $P<0.05$ was used for tests of statistical significance.

### Ethical considerations

The study protocol was reviewed and approved by the Institutional Review Board of the College of Health Sciences at Addis Ababa University (AAU) (protocol number: 042/20/SPH). Permission to undertake this study was obtained from every relevant authority at all levels. Official letters from AAU were written to each hospital to cooperate and participate in the survey. The purpose and significance of the study was introduced to the study participants, and all participants provided written or oral consent before participating in the study. Anonymity and data confidentiality were ensured, and no identifiable data from participants were collected.

## Results

### Characteristics of study participants

A total of 1,134 (92%) HCPs consented and completed the questionnaires, out of 1,228 possible participants. Among 1,134 healthcare personnel, nearly 40% of them were nurses, followed by physicians (22.4%) and intern doctors (10.8%). Table 1 summarizes the demographic and occupational characteristics of the study participants and their professional affiliation. About 47% were males, with females making 52.6% of the respondents. Among 982 participants with available data on age, the mean (±SD) age was 30.3±6.4 years and ranged from 22 to 70 years old, with the majority within the age group of 20–29 years (57.9%).

Among 252 physicians participated in the study, general practitioners and resident doctors accounted for 44.8% and 42.9%, respectively, while medical specialists and sub-specialists accounted for the remaining 12.3%. About 17% of the respondents represented other professional categories such as anesthetist, pharmacist, health officer, radiographer and laboratory technologist (Table 1). Majority (17.2%) of the respondents worked in obstetrics and gynecology department, while 13.8% were in surgical department, 13.3% in pediatrics, 13.0% in medical and 10.5% in OPD departments. Most respondents worked as staff for less than 10 years in the hospital (73.2%), and nearly 10% worked for 10 or more years.

### COVID-19 preventive practices

The self-reported prevalence of different preventive measures practiced by HCPs to prevent themselves from coronavirus infection is shown in Table 2. The overall highest practice showed among healthcare participants were wearing facemask (93%), hand washing for at least 20 seconds (92.7%), covering mouth and nose when coughing or sneezing (90.9%), and

**Table 1. Characteristics of study participants by professional category (N = 1,134).**

| Characteristics | Professional category, n (%) | | | | | Total, n (%) |
|---|---|---|---|---|---|---|
| | *Physician* | *Intern* | *Nurse* | *Midwife* | *Other*[a] | |
| **Gender (n = 1,134)** | | | | | | |
| Male | 157 (62.3) | 58 (47.2) | 175 (38.6) | 44 (37.6) | 103 (54.5) | 537 (47.4) |
| Female | 95 (37.7) | 65 (52.8) | 278 (61.4) | 73 (62.4) | 86 (45.6) | 597 (52.6) |
| **Age group (years) (n = 982)** | | | | | | |
| 20–29 | 101 (45.9) | 99 (91.7) | 220 (57.0) | 80 (79.2) | 69 (41.3) | 569 (57.9) |
| 30–39 | 106 (48.2) | 8 (7.4) | 119 (30.8) | 14 (13.9) | 70 (41.9) | 317 (32.3) |
| ≥40 | 13 (5.9) | 1 (0.9) | 47 (12.2) | 7 (6.9) | 28 (16.8) | 96 (9.8) |
| Mean (±SD) | 31.0 (±5.6) | 25.6 (±3.3) | 30.7 (±6.5) | 28.3 (±5.7) | 32.6 (±7.5) | 30.3 (±6.4) |
| Median (Range) | 30.0 (22–70) | 25.6 (22–45) | 30.7 (22.57) | 28.3 (22–52) | 32.3 (23–60) | 30.3 (22–70) |
| **Department/Unit (n = 1,134)** | | | | | | |
| Gyn&Ob | 27 (10.7) | 31 (25.2) | 36 (7.9) | 97 (82.9) | 4 (2.1) | 195 (17.2) |
| Surgical | 43 (17.1) | 31 (25.2) | 65 (14.3) | 2 (1.7) | 16 (8.5) | 157 (13.8) |
| Pediatrics | 39 (15.5) | 35 (28.5) | 71 (15.7) | 2 (1.7) | 4 (2.1) | 151 (13.3) |
| Medical | 62 (24.6) | 17 (13.8) | 62 (13.7) | 0.0 | 6 (3.2) | 147 (13.0) |
| OPD/Screening/Triage | 16 (6.3) | 2 (1.6) | 83 (18.3) | 6 (5.1) | 37 (19.6) | 144 (12.7) |
| Emergency | 28 (11.1) | 4 (3.3) | 34 (7.5) | 10 (8.5) | 19 (10.1) | 95 (8.4) |
| Anesthesia/OR/ICU | 12 (4.8) | 1 (0.8) | 66 (14.6) | 0.0 | 14 (7.4) | 93 (8.2) |
| Other[b] | 25 (9.9) | 2 (1.6) | 36 (7.9) | 0.0 | 89 (47.1) | 152 (13.4) |
| **Hospital (n = 1,134)[c]** | | | | | | |
| TASH | 79 (31.3) | 17 (13.8) | 128 (28.3) | 19 (16.2) | 40 (21.2) | 283 (25.0) |
| ZMH | 39 (15.5) | 36 (29.3) | 54 (11.9) | 15 (12.8) | 33 (17.5) | 177 (15.6) |
| GMH | 17 (6.7) | 7 (5.7) | 51 (11.3) | 21 (17.9) | 19 (10.1) | 115 (10.1) |
| Y12HMC | 35 (13.9) | 12 (9.8) | 48 (10.6) | 15 (12.8) | 42 (22.2) | 152 (13.4) |
| MH | 39 (15.5) | 29 (23.6) | 68 (15.0) | 20 (17.1) | 18 (9.5) | 174 (15.3) |
| SPHMMC | 43 (17.1) | 22 (17.9) | 104 (23.0) | 27 (23.1) | 37 (19.6) | 233 (20.5) |
| **Work experience (n = 938)** | | | | | | |
| <5 | 167 (79.5) | 84 (90.3) | 168 (44.0) | 65 (67.0) | 68 (43.6) | 552 (58.8) |
| 5–9 | 33 (15.7) | 7 (7.5) | 160 (41.9) | 25 (25.8) | 53 (34.0) | 278 (29.6) |
| 10–14 | 5 (2.4) | 2 (2.2) | 29 (7.6) | 4 (4.1) | 21 (13.5) | 61 (6.5) |
| 15–34 | 15 (2.4) | 0.0 | 25 (6.5) | 3 (3.1) | 14 (9.0) | 47 (5.0) |
| **Total, n (%)** | **252 (22.2)** | **123 (10.8)** | **453 (39.3)** | **117 (10.3)** | **189 (16.7)** | **1,134 (100)** |

[a]Other: Includes anesthetist, pharmacist, health officer, lab technologist and radiographer.

[b]Other: Includes Isolation room/ward, Pharmacy, Oncology, etc.

[c]TASH: Tikur Anbessa Specialized Hospital; ZMH: Zewditu Memorial Hospital; GMH: Ghandi Memorial Hospital; Y12HMC: Yekatit 12 Hospital Medical College; MH: Menelik II Hospital; SPHMMC: St. Paul Hospital Millennium Medical College.

avoiding touching eyes, nose, and mouth with unwashed hands (90.5%). These measures were commonly reported (>90%) for physicians, intern doctors, nurses and other HCPs except the midwives who reported <90%. A lower percentage of self-reported practices were observed in physical distancing (84.3%), the use of disinfecting surfaces (76.1%), and staying home when feeling cold or sick (64.6%), with similar pattern across the different categories of healthcare workers.

This study also investigated the attitude of the healthcare workers with regard to which group of people they recommend to use a facemask or N95 respirator. The majority of the respondents (94.8%) recommended the use of a facemask by all HCPs, all healthy people to

**Table 2. Self-reported prevalence of preventive measures practiced by healthcare professionals to prevent coronavirus infection by professional category (N = 1,134).**

| Variable | Professional category, % | | | | | Total, |
|---|---|---|---|---|---|---|
| | *Physician* | *Intern* | *Nurse* | *Midwife* | *Other[a]* | |
| | *(n = 252)* | *(n = 123)* | *(n = 453)* | *(n = 117)* | *(n = 189)* | *%(N = 1,134)* |
| Wearing facemask | 95.6 | 95.9 | 90.9 | 89.7 | 94.7 | 93.0 |
| Hand washing for at least 20 seconds | 95.2 | 95.1 | 90.9 | 88.9 | 94.2 | 92.7 |
| Covering your mouth and nose when you cough or sneeze | 93.7 | 96.7 | 89.0 | 87.2 | 90.5 | 90.9 |
| Avoiding touching your eyes, nose, and mouth with unwashed hands | 90.9 | 92.7 | 90.1 | 88.0 | 91.0 | 90.5 |
| Use of disinfectants to clean hands when water and soap was not available for washing hands | 92.9 | 93.5 | 83.9 | 83.8 | 90.5 | 88.0 |
| Physical distancing | 84.1 | 85.4 | 85.9 | 79.5 | 83.1 | 84.3 |
| Disinfecting mobile phone | 84.2 | 82.1 | 83.4 | 84.6 | 83.6 | 83.6 |
| Disinfecting surfaces | 73.0 | 73.2 | 79.0 | 74.4 | 76.2 | 76.1 |
| Staying home when you were sick or when you had a cold | 63.1 | 65.9 | 66.9 | 61.5 | 62.4 | 64.6 |

[a]Other: Includes anesthetist, pharmacist, health officer, lab technologist and radiographer.

protect themselves from coronavirus infection (90.1%), and people with close contact with suspected or confirmed COVID-19 (88.8%). About 87% of all respondents suggested that N95 respirator should be used by all HCPs as well as by people who are being in close contact with suspected or confirmed COVID-19 patients. About five in 10 (48%) of the respondents recommended the use of N95 respirator by healthy people to protect themselves against coronavirus infection. About 65% and 48% of the respondents from TASH and SPHMMC, respectively, recommended the use of N95 respirator for all healthy people to protect themselves from COVID-19.

## Preparedness for providing care to COVID-19 patients

Only about one-third (30.7%) of the study participated reported that they ever participated in direct clinical care to patients affected by infectious disease outbreaks such as Ebola, SARS and cholera. Nearly three in 10 (28.9%) respondents reported that they ever provided direct clinical care to at least one suspected/confirmed COVID-19 patient, with 39.1% participants from SPHMMC, 34.5% from MH and 31.1% from TASH. Regarding the level of preparedness of HCPs to provide direct clinical care to COVID-19 patients, 33.6% reported that they were prepared to provide direct clinical care to COVID-19 patients. In contrast, about two-third (66.4%) of the healthcare workers reported that they were not prepared to manage COVID-19 patients.

## Perceived risk of COVID-19

The study participants were asked three questions about their personal health, potential risks of becoming infected with COVID-19 or the potential risks to their families and loved ones due to their clinical role in the hospital. About 30% and 43% of the participants somewhat or strongly worried, respectively, that their personal health was at risk during the COVID-19 pandemic due to their role in the hospital (Table 3). Nevertheless, 6% and 13.5% of respondents reported that they somewhat not worried or even not worried at all that their personal health was not at risk due to COVID-19, respectively. About 38% and 50% of all respondents perceived that they were somewhat worried or extremely worried about themselves, respectively, due to the potential risk of becoming infected with coronavirus by their clinical role in the hospital setting, with only 5.6% perceived that they were not worried about the risk of being

**Table 3. Healthcare professional's worry about their clinical role in the hospital during COVID-19 by professional category.**

| Variable | Professional category, % | | | | | Total, % |
|---|---|---|---|---|---|---|
| | *Physician* | *Intern* | *Nurse* | *Midwife* | *Other*[a] | |
| | *(n = 244)* | *(n = 120)* | *(n = 431)* | *(n = 108)* | *(n = 181)* | *(n = 1,084)* |
| **How worried are you about your personal health due to your role in the hospital during COVID-19 pandemic?** | | | | | | |
| Extremely worried | 47.1 | 50.0 | 39.7 | 40.7 | 42.0 | 43.0 |
| Somewhat worried | 35.2 | 27.5 | 25.5 | 28.7 | 37.0 | 30.2 |
| Neutral | 4.9 | 8.3 | 9.5 | 6.5 | 5.5 | 7.4 |
| Somewhat not worried | 3.7 | 5.0 | 7.9 | 7.4 | 4.4 | 6.0 |
| Not worried at all | 9.0 | 9.2 | 17.4 | 16.4 | 11.0 | 13.5 |
| **How worried are you about the potential risk of becoming infected with COVID-19 due to your role in the hospital?** | | | | | | |
| Extremely worried | 47.1 | 56.7 | 48.5 | 58.3 | 46.4 | 49.7 |
| Somewhat worried | 47.5 | 35.0 | 34.8 | 29.6 | 40.3 | 38.1 |
| Neutral | 3.3 | 6.7 | 8.6 | 5.6 | 6.6 | 6.5 |
| Somewhat not worried | 2.0 | 1.7 | 5.1 | 2.8 | 4.4 | 3.7 |
| Not worried at all | 0.0 | 0.0 | 3.0 | 3.7 | 2.2 | 1.9 |
| **How worried are you about the potential risk of COVID-19 to your family, loved ones or others due to your role in the hospital?** | | | | | | |
| Extremely worried | 66.8 | 75.8 | 61.9 | 63.0 | 60.2 | 64.4 |
| Somewhat worried | 29.5 | 19.2 | 25.5 | 28.7 | 29.3 | 26.7 |
| Neutral | 2.5 | 4.2 | 7.4 | 4.6 | 5.0 | 5.3 |
| Somewhat not worried | 0.4 | 0.8 | 3.2 | 2.8 | 2.8 | 2.2 |
| Not worried at all | 0.8 | 0.0 | 1.9 | 0.9 | 2.8 | 1.5 |

[a]Other: Includes anesthetist, pharmacist, health officer, lab technologist and radiographer.

infected with the virus. Majorities of the respondents (64.4%) extremely worried about the potential risk of infection to their family and loved ones, and the remaining 26.7% were somewhat worried. Only 4.4% of the respondents were not worried about the risk of COVID-19 to their family and loved ones.

## Worry about COVID-19 crisis

Of the total 1,134 study participants, 952 (84%) had complete responses on all the 12-items for computing the total score of worry about the current COVID-19 crisis. About 66% of the respondents reported that they worried a lot about losing someone due to COVID-19, 66.7% worried a lot about the health of their loved ones, and 67.5% worried a lot about the health system being overloaded by the patients of COVID-19, followed by a lot of worries about the economic recession in the country (58%) and restricted access to food supplies (56.1%) (Table 4). The study also revealed that there were respondents who were ambivalent or didn't worry at all about COVID-19 crisis.

The total score of the worry scale about COVID-19 crisis was calculated by summing up the score of the 12 questions. The higher the score, the greater the worry of the COVID-19 crisis. Table 5 presents the mean scores for each item and the overall worry scores of the COVID-19 crisis by professional category. Overall, the participants reported an average of moderate-to-high levels of worry about COVID-19 crisis on each item (2.37), ranging from 2.1 on 'becoming unemployed' to 2.6 on 'losing someone they love', 'health system being overloaded' and 'someone's loved health'.

**Table 4. Healthcare professional's worry about COVID-19 crisis by professional category (N = 12 items).**

| Worry questions | Professional category, % | | | | | Total, % |
|---|---|---|---|---|---|---|
| | *Physician* | *Intern* | *Nurse* | *Midwife* | *Other[a]* | |
| | *(n = 221)* | *(n = 110)* | *(n = 374)* | *(n = 95)* | *(n = 152)* | *(n = 952)* |
| **Loosing loved one's** | | | | | | |
| Don't worry at all | 7.2 | 10.0 | 12.6 | 15.8 | 12.5 | 11.3 |
| Worry somehow | 25.3 | 19.1 | 21.1 | 22.1 | 23.7 | 22.4 |
| Worry a lot | 67.4 | 70.4 | 66.3 | 62.1 | 63.8 | 66.3 |
| **Health system being overwhelmed** | | | | | | |
| Don't worry at all | 7.7 | 5.5 | 8.8 | 9.5 | 9.2 | 8.3 |
| Worry somehow | 17.2 | 30.0 | 24.1 | 29.5 | 27.0 | 24.2 |
| Worry a lot | 75.1 | 64.5 | 67.1 | 61.1 | 63.8 | 67.5 |
| **Own mental health** | | | | | | |
| Don't worry at all | 19.9 | 30.0 | 22.2 | 21.1 | 25.0 | 22.9 |
| Worry somehow | 44.8 | 30.9 | 35.6 | 37.9 | 36.2 | 37.5 |
| Worry a lot | 35.3 | 39.1 | 42.2 | 41.1 | 38.8 | 39.6 |
| **Own physical health** | | | | | | |
| Don't worry at all | 11.8 | 12.7 | 17.4 | 11.6 | 17.8 | 15.0 |
| Worry somehow | 45.2 | 40.9 | 38.8 | 45.3 | 38.2 | 41.1 |
| Worry a lot | 43.0 | 46.4 | 43.9 | 43.2 | 44.1 | 43.9 |
| **Loved ones' health** | | | | | | |
| Don't worry at all | 12.7 | 8.2 | 10.2 | 10.5 | 9.2 | 10.4 |
| Worry somehow | 21.3 | 14.5 | 26.5 | 26.3 | 20.4 | 22.9 |
| Worry a lot | 66.1 | 77.3 | 63.4 | 63.2 | 70.4 | 66.7 |
| **Restricted liberty of movement** | | | | | | |
| Don't worry at all | 13.6 | 18.2 | 12.8 | 13.7 | 13.2 | 13.8 |
| Worry somehow | 44.8 | 41.8 | 43.9 | 43.2 | 49.3 | 44.6 |
| Worry a lot | 41.6 | 40.0 | 43.3 | 43.2 | 37.5 | 41.6 |
| **Companies running out of business** | | | | | | |
| Don't worry at all | 10.9 | 13.6 | 14.2 | 16.8 | 12.5 | 13.3 |
| Worry somehow | 50.2 | 50.0 | 37.2 | 35.8 | 38.2 | 41.7 |
| Worry a lot | 38.9 | 36.4 | 48.7 | 47.4 | 49.3 | 45.0 |
| **Economic recession** | | | | | | |
| Don't worry at all | 7.7 | 7.3 | 9.1 | 7.4 | 10.5 | 8.6 |
| Worry somehow | 37.1 | 47.3 | 29.7 | 36.8 | 25.0 | 33.4 |
| Worry a lot | 55.2 | 45.5 | 61.2 | 55.8 | 64.5 | 58.0 |
| **Restricted access to food supplies** | | | | | | |
| Don't worry at all | 11.3 | 5.5 | 9.9 | 6.3 | 8.6 | 9.1 |
| Worry somehow | 37.1 | 35.5 | 31.6 | 36.8 | 37.5 | 34.8 |
| Worry a lot | 51.6 | 59.1 | 58.6 | 56.8 | 53.9 | 56.1 |
| **Becoming unemployed** | | | | | | |
| Don't worry at all | 51.1 | 27.3 | 25.4 | 22.1 | 28.9 | 31.8 |
| Worry somehow | 19.5 | 27.3 | 32.9 | 32.6 | 26.3 | 28.0 |
| Worry a lot | 29.4 | 45.5 | 41.7 | 45.3 | 44.7 | 40.1 |
| **Inability to pay my bills** | | | | | | |
| Don't worry at all | 30.8 | 23.6 | 18.2 | 17.9 | 17.1 | 21.5 |
| Worry somehow | 37.1 | 33.6 | 42.2 | 43.2 | 41.4 | 40.0 |
| Worry a lot | 32.1 | 42.7 | 39.6 | 38.9 | 41.4 | 38.4 |
| **Inability to visit relatives** | | | | | | |

*(Continued)*

**Table 4.** (Continued)

| Worry questions | Professional category, % | | | | | Total, % |
|---|---|---|---|---|---|---|
| | *Physician* | *Intern* | *Nurse* | *Midwife* | *Other*[a] | |
| | *(n = 221)* | *(n = 110)* | *(n = 374)* | *(n = 95)* | *(n = 152)* | *(n = 952)* |
| Don't worry at all | 10.9 | 14.5 | 8.8 | 4.2 | 12.5 | 10.1 |
| Worry somehow | 32.6 | 29.1 | 34.8 | 40.0 | 27.6 | 33.0 |
| Worry a lot | 56.6 | 56.4 | 56.4 | 55.8 | 59.9 | 56.9 |

[a]Other: Includes anesthetist, pharmacist, health officer, lab technologist and radiographer.

The overall average worry scores of the 12 items for the COVID-19 crisis was high, with a mean (±SD) of 28.4 (±5.9), ranging from 12 to 36. The total average worry scores for the hospitals ranged from 25.6 (±6.8) at TASH to 31.3 (±5.0) at GMH; and was further categorized into three levels (i.e., low, moderate, and high). Fig 1 shows the pattern of the total worry score of COVID-19 crisis, and about 56% of respondents from TASH showed a relatively low worry score compared to the highest (50.9%) worry score reported by participants from GMH.

Table 6 shows the standardized regression coefficients of predictors of overall worry scores of HCPs about the COVID-19 crisis. The predictors gender, professional category, and department/unit did not significantly predict worry about the COVID-19 crisis. Similarly, no significant differences were found between HCPs who had provided clinical care to suspected/confirmed COVID-19 patients or those who were prepared to provide direct clinical care to COVID-19 patients in terms of worry about COVID-19 crisis than those who had not. Whereas, participants who had previously provided clinical care to Ebola, SARS and cholera patients had significantly lower levels of worry about COVID-19 crisis than participants who had not ($\beta$ = -1.38, $P$<0.001). In addition, HCPs from TASH ($\beta$ = -3.38, $P$<0.001) and Y12HMC ($\beta$ = -1.31, $P$ = 0.043) had significantly lower levels of worry about COVID-19 crisis

**Table 5. Mean worry score of healthcare professionals about COVID-19 crisis by professional category (N = 12 items).**

| Worry question | Professional category, Mean (SD)[a] | | | | | Mean (SD) |
|---|---|---|---|---|---|---|
| | *Physician* | *Intern* | *Nurse* | *Midwife* | *Other*[b] | |
| | *(n = 221)* | *(n = 110)* | *(n = 374)* | *(n = 95)* | *(n = 152)* | *(n = 952)* |
| Losing loved one's | 2.6 (0.6) | 2.6 (0.7) | 2.5 (0.7) | 2.5 (0.8) | 2.5 (0.7) | 2.6 (0.7) |
| Health system being overwhelmed | 2.7 (0.6) | 2.6 (0.6) | 2.6 (0.6) | 2.5 (0.7) | 2.6 (0.7) | 2.6 (0.6) |
| Own mental health | 2.2 (0.7) | 2.1 (0.8) | 2.2 (0.8) | 2.2 (0.8) | 2.1 (0.8) | 2.2 (0.8) |
| Own physical health | 2.3 (0.7) | 2.3 (0.7) | 2.3 (0.7) | 2.3 (0.7) | 2.3 (0.7) | 2.3 (0.7) |
| Loved one's health | 2.5 (0.7) | 2.7 (0.6) | 2.5 (0.7) | 2.5 (0.7) | 2.6 (0.7) | 2.6 (0.7) |
| Restricted liberty of movement | 2.7 (0.7) | 2.2 (0.7) | 2.3 (0.7) | 2.3 (0.7) | 2.2 (0.7) | 2.3 (0.7) |
| Companies running out of business | 2.3 (0.7) | 2.2 (0.7) | 2.4 (0.7) | 2.3 (0.7) | 2.4 (0.7) | 2.3 (0.7) |
| Economic recession | 2.5 (0.6) | 2.4 (0.6) | 2.5 (0.7) | 2.5 (0.6) | 2.5 (0.7) | 2.5 (0.7) |
| Restricted access to food supplies | 2.4 (0.7) | 2.5 (0.6) | 2.5 (0.7) | 2.5 (0.6) | 2.6 (0.6) | 2.5 (0.7) |
| Becoming unemployed | 1.8 (0.9) | 2.2 (0.8) | 2.2 (0.8) | 2.2 (0.8) | 2.2 (0.8) | 2.1 (0.8) |
| Inability to pay my bills | 2.0 (0.8) | 2.2 (0.8) | 2.2 (0.7) | 2.2 (0.7) | 2.2 (0.7) | 2.2 (0.8) |
| Inability to visit relatives | 2.5 (0.6) | 2.4 (0.7) | 2.5 (0.7) | 2.5 (0.6) | 2.5 (0.7) | 2.5 (0.7) |
| **Overall mean (SD)** | **27.9 (5.9)** | **28.5 (5.6)** | **28.7 (6.1)** | **28.6 (5.8)** | **28.6 (5.7)** | **28.4 (5.9)** |

[a]Numbers in parentheses represent standard deviations.
[b]Other: Includes anesthetist, pharmacist, health officer, lab technologist and radiographer.

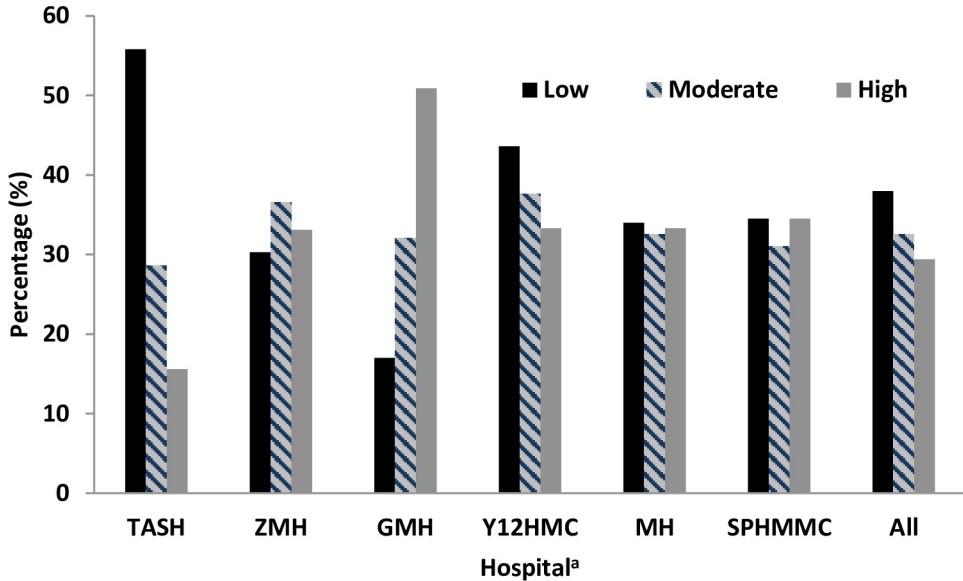

**Fig 1. Pattern of worry scores of COVID-19 crisis by hospital.** [a]TASH: Tikur Anbessa Specialized Hospital; ZMH: Zewditu Memorial Hospital; GMH: Ghandi Memorial Hospital; Y12HMC: Yekatit 12 Hospital Medical College; MH: Menelik II Hospital; SPHMMC: St. Paul Hospital Millennium Medical College.

than those participants from the SPMMC, whereas participants from GMH (β = 1.64, *P* = 0.021) had significantly higher levels of worry about COVID-19.

## Discussion

Since its emergence in December 2020, the COVID-19 pandemic is a global public health concern and the most current topic of discussion across every facet of life, especially among the HCPs and patients. This study was conducted in Addis Ababa city between June 09 and 26, 2020. This was three months after the first confirmed case of COVID-19 was reported in Ethiopia. Addis Ababa city is the most affected part in the country, and the study was conducted when strict measures were taken to contain the COVID-19 pandemic. The study aimed to assess protective behaviors, risk perceptions, and worry about the COVID-19 crisis among HCPs in six public hospitals the city. The study participants included medical doctors, interns, nurses, midwives, pharmacists, medical laboratory technologists, and technicians. These categories of HCPs have direct or indirect close personal exposures with suspected or confirmed COVID-19 patients while performing their clinical duties. The study revealed widespread practices of preventive measures, the highest perceived risk and worry about COVID-19 crisis among healthcare workers.

The majority of the participants in the study reported a high level of practice towards the prevention of SARS-CoV-2 infection particularly regarding using facemask, hand washing for at least 20 seconds, covering mouth and nose when coughing or sneezing, and avoiding touching eyes, nose, and mouth with unwashed hands as far as possible. This finding is consistent with the finding of a similar study conducted in China, where the potential risk of COVID-19 has largely improved the infection prevention and control behaviors of HCPs working in hospitals [18]. In a study conducted in Egypt, hand washing, refraining from touching eyes, mouth and nose, and using surgical facemask were the most frequently accepted preventive measures among health workers [19]. The WHO recommends the use of primary preventive

**Table 6. Standardized linear regression beta-coefficients (β) of predictors associated with worry scores of COVID-19 crisis (n = 952).**

| Predictor | β (95% CI)[a] | SE | t-statistic | *P*-value |
|---|---|---|---|---|
| **Gender (Male)** | -0.29 (-1.03, -0.46) | 0.38 | -0.75 | 0.451 |
| **Professional category** | | | | |
| Physician | 1[e] | | | |
| Intern | -0.89 (-2.23, -0.46) | 0.69 | -1.29 | 0.196 |
| Nurse | -0.20 (-1.18, -0.79) | 0.50 | -0.39 | 0.696 |
| Midwife | -1.12 (-2.71, 0.48) | 0.82 | -1.37 | 0.171 |
| Other[b] | -0.24 (-1.54, -1.07) | 0.67 | -0.35 | 0.724 |
| **Department/Unit** | | | | |
| Gyn&Ob | 1 | | | |
| Surgical | -0.84 (-2.37, -0.68) | 0.78 | -1.08 | 0.279 |
| Pediatrics | -0.43 (-1.93, 1.06) | 0.76 | -0.57 | 0.569 |
| Medical | -1.82 (-3.37, -0.27) | 0.79 | -2.30 | **0.022** |
| OPD/Screening/Triage | -0.85 (-2.39, 0.69) | 0.79 | -1.08 | 0.279 |
| Emergency | -1.36 (-3.01, 0.28) | 0.84 | -1.63 | 0.104 |
| Anesthesia/OR/IC | 1.28 (-0.47, 3.02) | 0.89 | 1.44 | 0.151 |
| Other[c] | -0.93 (-2.59, 0.74) | 0.85 | -1.09 | 0.274 |
| **Hospital[d]** | | | | |
| SPHMMC | 1 | | | |
| TASH | -3.38 (-4.50, -2.26) | 0.57 | -5.94 | **<0.001** |
| ZMH | 0.16 (-1.08, 1.40) | 0.63 | 0.26 | 0.795 |
| GMH | 1.64 (0.25, 3.04) | 0.71 | 2.31 | **0.021** |
| Y12HMC | -1.31 (-2.58, -0.04) | 0.65 | -2.03 | **0.043** |
| MH | 0.06 (-1.15, 1.27) | 0.62 | 0.10 | 0.924 |
| **Prepared to provide direct care to COVID-19 cases** | 0.33 (-046, 1.12) | 0.40 | 0.83 | 0.409 |
| **Ever provided clinical care to suspected/ confirmed COVID-19 patients** | -0.29 (-1.12, 0.53) | 0.42 | -0.69 | 0.488 |
| **Ever provided clinical care to Ebola, SARS and cholera patients** | -1.38 (-2.21, -0.55) | 0.42 | -3.28 | **<0.001** |

[a]CI: Confidence Interval, SE: Standard Error.

[b]Other: Includes anesthetist, pharmacist, health officer, lab technologist and radiographer.

[c]Other: Includes Isolation room/ward, Pharmacy, Oncology, etc.

[d]TASH: Tikur Anbessa Specialized Hospital; ZMH: Zewditu Memorial Hospital; GMH: Ghandi Memorial Hospital; Y12HMC: Yekatit 12 Hospital Medical College; MH: Menelik II Hospital; SPHMMC: St. Paul Hospital Millennium Medical College.

[e]Reference.

measures that includes regular hand washing, physical distancing, and respiratory hygiene (covering mouth and nose while coughing or sneezing) by healthcare workers in order to prevent the spread of the virus among themselves and patient's close contacts [20].

Studies conducted during the early stage of the pandemic revealed that healthcare workers had insufficient knowledge about COVID-19 pandemic to protect themselves from coronavirus infection [21]. In a study conducted in Greece, only 25% of healthcare practitioners washed their hands after touching a patient, despite the fact that 94% of the respondents knew that SARS-CoV-2 transmission could be reduced with hand washing [22]. Although hand washing is recommended for the general public in order to prevent the transmission of COVID-19, hand hygiene is mandatory for healthcare practitioners, in order to prevent infections, both for oneself and for the patients [23]. In the present study, the use of facemask was reported to be 93%. A recent study conducted in Addis Ababa just before the current study revealed that about two-third of the healthcare workers demonstrated a poor practice of facemask utilization

[24]. Similar results were reported in India that majority of the healthcare workers (91%) reported that they used surgical masks, 97% of the participants used hand sanitizers and 97% practiced proper hand hygiene [25].

In the present study, the majority of the study participants recommended mask-wearing for all HCPs, all healthy people to protect themselves from coronavirus infection, and people with close contact with suspected or confirmed COVID-19. Similarly, about 87% of the respondents suggested that N95 respirator should be used by all HCPs as well as by people who are being in close contact with suspected or confirmed COVID-19 patients. In Pakistan, 71% of the health-care workers believed that wearing general medical masks was protective against COVID-19 [26], and studies also suggested that surgical masks are similarly as effective as N95 respirators if used with hand wash and other infection prevention precautions [27]. However, a rapid systematic review on the efficacy of facemasks and respirators against coronaviruses and other respiratory transmissible viruses reported that continuous use of respirators is more protective compared to the medical masks, and medical masks are more protective than cloth masks among health workers in healthcare settings [28].

This study demonstrated that about one-third of all respondents in the study either participated in direct clinical care to patients affected by an infectious disease outbreak (e.g., Ebola virus, SARS, cholera, Zika virus) or provided direct clinical care at least for one suspected or confirmed COVID-19 patients during the current COVID-19 epidemic. This percentage is higher than the figures reported by other studies on this subject in the early days of the COVID-19 outbreak in China [29]. A significant number (38%) of HCPs in the current study expressed lack of or low level of preparedness to manage suspected or confirmed COVID-19 patients. This raises a concern regarding the ability and confidence of the healthcare workers to combat COVID-19 infection. Despite these concerns, along with the shortage of PPE and inadequate training during the COVID-19 pandemic, the healthcare workers continue to work with the management of suspected or confirmed COVID-19, working in the hospital setting where COVID-19 patients were admitted, risking their lives to save their patients. However, this could highlight the risk of infection among healthcare workers and cross-contamination within hospitals and could lead to a higher rate of hospital-acquired infections. Therefore, the study provides considerable insights into the necessity of immediate and determined efforts focused on training programs and providing an adequate supply of PPE to ensure the safety of health personnel during the COVID-19 pandemic [30].

In the present study, about 88% of the HCPs were afraid of being infected with SARS-CoV-2 and about 91% worried about the potential risk of transmitting the virus to their family and loved ones. The risk of contracting the virus was perceived to be very high at the time of the study. The HCPs expressed worry and fear of infection due to the contagious nature of the virus, their close contact with suspected or confirmed COVID-19 patients, and taking infection home to their family and colleagues. In Iran, it was found that about 92% of the healthcare workers worried about being infected with the virus and transmitting it to the family [31]. In a study conducted in Henan province of China, 89% of healthcare workers had sufficient knowledge of COVID-19, 85% were concerned about infection with the virus, and 90% followed correct practices regarding the prevention of COVID-19 [32]. About 83% of the healthcare workers in Egypt reported increased risk perception because of the concern of being infected with COVID-19 and fear of transmitting the disease to their families, and 89% stated that they were more susceptible to SARS-CoV-2 infection mainly due to the shortage of appropriate PPE [19].

In the current study, the overall worry scores of the study participants regarding the COVID-19 crisis was considerably higher, exposure to previous outbreaks/pandemics such as involvement in the clinical care of Ebola, SARS, and cholera patients predicts a significantly

lower level of worry about the COVID-19 crisis. One possible reason for this result is the healthcare workers with previous experience in the management of outbreak/epidemic are adapted to such emergencies, which might be associated with lower degree of worries. However, there was no significance difference among the different categories of the HCPs in the degree of worry. Various studies have reported the psychological impact of COVID-19 on HCPs [33]. A recent review found that the frontline healthcare workers are at an increased risk of direct physical and mental consequences as the result of providing care to COVID-19 patients [34]. Studies demonstrated that more than half of the HCPs report symptoms of depression, insomnia, and anxiety due to COVID-19 [35].

A recent study carried out in Pakistan reported that about three-fourth of HCPs had fear of getting infected during the management of COVID-19 patients, and another two-third reported severe anxiety, which was particularly more common among nurses [36]. Studies also reported excessive workload, isolation, mental stress and discrimination among frontline health professionals, thus, contributing to worry and fear, physical exhaustion, and emotional disturbance [37]. A Cochrane review reported the suffering of healthcare workers from work-related or occupational stress, which can be reduced by cognitive-behavioral training as well as mental and physical relaxation [38]. A multicenter study conducted among frontline nurses in China showed poor mental health during the COVID-19 outbreak, mainly due to the fear of contracting the virus and high workload [39]. Moreover, the same study revealed that nurses who were confident in their infection prevention knowledge and skills had lower stress levels than those who felt less prepared.

Overall, the participants of the present study indicated higher perceived risk and worry about the COVID-19 crisis, which could ultimately affect motivation and performance related to their clinical practice, particularly treating COVID-19 patients. Although the prevalence of preventive measures among the respondents was very high, active interventions such as the provision of adequate PPE and psychological support for HCPs should be considered not only for frontline healthcare workers, but also for all categories of health workers.

Finally, this study had several limitations. First, the study had a potential to be affected by selection bias and eligible participants might be excluded. Second, this study was conducted in six public hospitals in Addis Ababa, and may possibly limit the generalization of the results and findings to other public and private hospitals. Third, the study focused on more general populations of HCPs similar to other studies [32, 40] rather than healthcare workers who might have direct contact with COVID-19 patients [41]. At last, the results of this study might be affected by information bias since it was based on self-reported data using self-administered questionnaire, and the respondents might overestimate or underestimate the responses in a way that they believe is socially acceptable rather than reporting actual or genuine answers. Despite these limitations, the results obtained provide important information to guide health communication efforts that can support prevention efforts of COVID-19 among HCPs.

## Conclusions

In conclusion, our study has illuminated the widespread practices of preventive measures, higher levels of perceived risk and worry about the COVID-19 crisis among HCPs who have direct or indirect contact with COVID-19 patients. The study showed a universally higher preventive practices among the study participants to prevent SARS-CoV-2 infections. The higher level of worry about COVID-19 might be helpful in improving the perceived risk of the pandemic among the HCPs, which can motivate them to adopt proper preventive measures. This can be addressed through the implementation of risk communication programs with the public and healthcare workers during the current COVID-19 pandemic.

## Supporting information

**S1 Dataset.**
(SAV)

## Acknowledgments

The authors are grateful to the participating hospitals and their healthcare staff for committing their time and voluntarily for contributing to the research. They are also thankful to all data collectors and logistics facilitators for their time and commitment.

## Author Contributions

**Conceptualization:** Wakgari Deressa, Alemayehu Worku, Workeabeba Abebe, Wondwossen Amogne.

**Data curation:** Wakgari Deressa.

**Formal analysis:** Wakgari Deressa, Alemayehu Worku.

**Funding acquisition:** Wakgari Deressa, Alemayehu Worku, Workeabeba Abebe, Wondwossen Amogne.

**Investigation:** Wakgari Deressa, Alemayehu Worku, Workeabeba Abebe, Muluken Gizaw, Wondwossen Amogne.

**Methodology:** Wakgari Deressa, Alemayehu Worku, Workeabeba Abebe, Muluken Gizaw, Wondwossen Amogne.

**Project administration:** Wakgari Deressa.

**Resources:** Wakgari Deressa, Alemayehu Worku.

**Supervision:** Alemayehu Worku, Muluken Gizaw.

**Validation:** Alemayehu Worku.

**Writing – original draft:** Wakgari Deressa, Alemayehu Worku, Workeabeba Abebe, Muluken Gizaw, Wondwossen Amogne.

**Writing – review & editing:** Wakgari Deressa, Alemayehu Worku, Workeabeba Abebe, Muluken Gizaw, Wondwossen Amogne.

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
