## [Decision Letter · Decision Letter 0]

15 Mar 2021

PONE-D-20-33821

Risk perceptions and preventive practices of COVID-19 among healthcare professionals in public hospitals in Ethiopia

PLOS ONE

Dear Author,

Thank you for submitting your manuscript to PLOS ONE. After careful consideration, we feel that it has merit but does not fully meet PLOS ONE’s publication criteria as it currently stands. Therefore, we invite you to submit a revised version of the manuscript that addresses the points raised during the review process.

We look forward to receiving your revised manuscript.

Kind regards,

Ramesh Kumar, PhD

Academic Editor

PLOS ONE

Journal Requirements:

Reviewers' comments:

Reviewer's Responses to Questions

**Comments to the Author**

1. Is the manuscript technically sound, and do the data support the conclusions?

Reviewer #1: Yes

Reviewer #2: Partly

2. Has the statistical analysis been performed appropriately and rigorously? 

Reviewer #1: I Don't Know

Reviewer #2: No

3. Have the authors made all data underlying the findings in their manuscript fully available?

Reviewer #1: Yes

Reviewer #2: Yes

4. Is the manuscript presented in an intelligible fashion and written in standard English?

Reviewer #1: Yes

Reviewer #2: No

5. Review Comments to the Author

Reviewer #1: In the manuscript entitled “Risk perceptions and preventive practices of COVID-19 among healthcare professionals in public hospitals in Ethiopia” Following modifications should be done prior to acceptance

Lines 120 – 127 should be restructured

Abstract must include few lines of scientific outcome

Conclusion and abstract should be revisited to make the coherence outcome.

Reviewer #2: Thank you for the opportunity to review this manuscript. These are some issues that need serious revision

This study is very interesting and need of the current era. But the author has represented it in very general way.

The abstract does not fully capture the scope of the article. The reasons laid out by the authors are not fully in line with the abstract. Please consider revising the abstract section.

Similar studies have already been reported by various other researchers in this field, therefore I didn’t find anything new in this MS.

What is your contribution to the existing literature/body of knowledge in considering this study in the study setting? [I.e. The new knowledge/findings gained from the study were NOT clearly discussed]

There is no description of the tool the researcher used [i.e. information on its psychometric characteristics e.g., reliability and validity].

How you control the bias associated with self-administered questioner

What were the data quality measures you considered in the study

The text has a lot of typo and grammatical mistakes. Authors should thoroughly check and improve the language of the manuscript. Some of the text needs to be revised to tone down the narrative. The errors are too numerous to be listed here.

Why you use logistic regression? I don’t agree with you about your analysis method because for this study liner regression is appropriate, why not you use this?

Please synthesize the result section specifically in characteristics of the study participants

The statistical analysis section (Table-6): Look at the value of the COR and AOR almost all the AOR>COR how this happened and do you trust this value. For me both sampling and systematic error are introduced in your result please see it once again. I have a serious dought

What is the policy implication of this study and what each finding it implies, please enlist in the discussion section

Concluding remarks are also not very convincing and require more careful evaluation by authors to reflect the message of this paper.

Implications for practice, research, and theory were NOT clearly discussed. –Recommendations are targeted to the stakeholders and founded on the findings gained from the study.

6. PLOS authors have the option to publish the peer review history of their article (what does this mean?). If published, this will include your full peer review and any attached files.

Reviewer #1: **Yes: **Dr. Muhammad Abaid Ullah

Reviewer #2: No

---

## [Author Response · Author response to Decision Letter 0]

14 May 2021

Academic Editor’s Comments:

Comment 1. Please ensure that your manuscript meets PLOS ONE's style requirements, including those for file naming. 

The PLOS ONE style templates can be found at

Authors’ response: Thank you for this comment, which we have addressed and fulfilled all the PLOS ONE’s style requirements. 

Comment 2. PLOS requires an ORCID iD for the corresponding author in Editorial Manager on papers submitted after December 6th, 2016. Please ensure that you have an ORCID iD and that it is validated in Editorial Manager. To do this, go to ‘Update my Information’ (in the upper left-hand corner of the main menu), and click on the Fetch/Validate link next to the ORCID field. This will take you to the ORCID site and allow you to create a new iD or authenticate a pre-existing iD in Editorial Manager. Please see the following video for instructions on linking an ORCID iD to your Editorial Manager account: https://www.youtube.com/watch?v=_xcclfuvtxQ

 Authors’ response: Thank you for this comment. The corresponding author has a pre-existing ORCID iD (https://orcid.org/0000-0002-9712-2375), but that was linked to another email (deressaw@yahoo.com) with another username (Amente2532) in the Editorial Manager system. We requested the editor via email to delete the duplicate user account in the Editorial Manager system. As a result, they combined both records into one record in the system and solved our problem.

Now we have successfully authenticated a pre-existing ORCID ID in the Editorial Manager system. 

Comment 3. Your ethics statement should only appear in the Methods section of your manuscript. If your ethics statement is written in any section besides the Methods, please delete it from any other section.

Authors’ response: Thank you for this comment. The ethics statement now only appears in the Methods section of the manuscript, and we removed the statement on ethics which was written in the other part of the manuscript.

Comment 4. We note that you have indicated that data from this study are available upon request. PLOS only allows data to be available upon request if there are legal or ethical restrictions on sharing data publicly. For more information on unacceptable data access restrictions, please see http://journals.plos.org/plosone/s/data-availability#loc-unacceptable-data-access-restrictions.

Author’s Response: After having discussions among the research team and with the IRB, we have decided to upload the dataset of this study along the revised manuscript (S1 dataset). Although we initially indicated that the data from this study would be available upon request, now we understood the importance of availing the data.

Reviewers' comments:

Reviewer's Responses to Questions

Comments to the Author

1. Is the manuscript technically sound, and do the data support the conclusions?

Reviewer #1: Yes

Reviewer #2: Partly

2. Has the statistical analysis been performed appropriately and rigorously? 

Reviewer #1: I Don't Know

Reviewer #2: No

 3. Have the authors made all data underlying the findings in their manuscript fully available?

Reviewer #1: Yes

Reviewer #2: Yes

4. Is the manuscript presented in an intelligible fashion and written in standard English?

Reviewer #1: Yes

Reviewer #2: No

 5. Review Comments to the Author

Reviewers' comments:

Reviewer #1: 

In the manuscript entitled “Risk perceptions and preventive practices of COVID-19 among healthcare professionals in public hospitals in Ethiopia” the following modifications should be done prior to acceptance.

Comment 1: Lines 120 – 127 should be restructured

Authors’ response: Thank you for this comment, and we have now restructured the Methods section as “Study design and setting, Study population and sample size calculation, Sampling procedures, and Survey instrument and data collection” and rewritten the statement 120-127 as follows:

“The study population included intern and resident doctors, general practitioners, medical specialists and sub-specialists, health officers, anesthetists, nurses, midwives, pharmacists, laboratory technicians and technologists, physiotherapists and X-ray technicians, all of whom may expect to encounter suspected or confirmed COVID-19 patients. The six study hospitals included: Tikur Anbessa Specialized Hospital (TASH), Zewditu Memorial Hospital (ZMH), Ghandi Memorial Hospital (GMH), Menelik II Hospital (MH), Yekatit 12 Hospital Medical College (Y12HMC) and St. Paul Hospital Millennium Medical College (SPHMMC).” (Lines 118 to 125, Page 6).

Comment 2: Abstract must include few lines of scientific outcome

Authors’ response: Thank you for the comments. We significantly revised the abstract accordingly and included the main findings of the study.

Comment 3: Conclusion and abstract should be revisited to make the coherence outcome.

Authors’ response: Thank you for the two comments. We revised the conclusion of the main text as follows:

“In conclusion, our study has illuminated the widespread practices of preventive measures, high levels of perceived risk, and worry about COVID-19 crisis among healthcare professionals who have direct or indirect contact with COVID-19 patients. The study also identified factors associated with worry related to the COVID-19 crisis. The higher level of worry about COVID-19 might help improve the perceived risk of the pandemic among healthcare professionals, which can motivate them to adopt proper preventive measures. This can be addressed by implementing risk communication programs with the public and healthcare workers during the current COVID-19 pandemic.” (Lines 500 to 507, Page 28).

Based on the above main Conclusion of the study, we have also revised the Conclusion part of the abstract as follows:

“Our findings reveal respondents’ widespread practice of preventive measures, higher levels of perceived risk and worry about COVID-19 crisis. The higher perceived risk and worry about COVID-19 might enable healthcare workers to adopt appropriate preventive measures more effectively against the disease.” (Lines 37 to 40, Page 2).

Reviewer #2: 

Thank you for the opportunity to review this manuscript. These are some issues that need serious revision. This study is very interesting and need of the current era. But the author has represented it in very general way.

Comment 1: The abstract does not fully capture the scope of the article. The reasons laid out by the authors are not fully in line with the abstract. Please consider revising the abstract section.

Similar studies have already been reported by various other researchers in this field, therefore I didn’t find anything new in this MS.

Authors’ response: Thank you for the comments. We significantly revised the abstract accordingly and further included the main findings of the study.

Comment 2: What is your contribution to the existing literature/body of knowledge in considering this study in the study setting? [I.e., The new knowledge/findings gained from the study were NOT clearly discussed]

Authors’ response: Thank you for these useful comments. We revised the last paragraph of the ‘Background’ section to reflect on the above question (Lines 89 to 99, Page 5) as follows:

“Since the first case of COVID-19 was reported in Ethiopia on 13 March 2020, the Ministry of Health, in collaboration with its partners, conducted different trainings on preventive measures for HCPs [healthcare professionals] at several hospitals and health centers, with supplies of PPE materials. The HCPs in Ethiopia have worked tirelessly and played a crucial role in managing COVID-19 cases, despite high personal risks and worries about the current pandemic crisis. However, no study has been undertaken in the country on risk perception and preventive practices of HCPs during the current COVID-19 pandemic. In addition, emotional reactions and feelings of healthcare workers, such as worries about the COVID-19 crisis, have not been studied. This study was conducted to assess preventive practices, perceived risk and worry about COVID-19 among HCPs. Understanding the preventive practices, risk perception, and worries of HCPs would help in protecting them, and preventing the COVID-19 pandemic through effective risk communication.”

Comment 3: There is no description of the tool the researcher used [i.e., information on its psychometric characteristics e.g., reliability and validity].

Authors’ response: Thank you for the comments on the data collection tools. We have restructured the issue as “Survey instrument and data collection” and revised the section by dividing it into two paragraphs, which read as follows, and we included three more references to this section (Lines 143 to 173, Pages 7-8):

“A structured paper-based self-administered questionnaire was used to collect the data. The questionnaire was developed in English by the authors of the study by reviewing the previously conducted studies [15, 16] and adopting frequently asked questions from the WHO website on prevention measures, risk perception and emotional reaction such as worry about the COVID-19 crisis [17]. The questionnaire comprises parts on the demographic (gender, age) and occupational characteristics of the respondents (hospital, department/unit, professional category, and work experience), their preventive practices and preparedness to combat COVID-19, risk perception and worry about the current COVID-19 crisis. Questions related to measures taken to prevent infection from coronavirus included hand washing for at least 20 seconds, using disinfectants, wearing a facemask, staying home, physical distancing, covering mouth, and nose while coughing and sneezing, and other preventive measures. Most of the questions were designed as ‘yes/no’. 

Risk perception in this study was measured using three questions: perceived worry about their health, perceived risk of becoming infected with coronavirus, and the potential risk to their family, loved ones or others due to their role in the hospital. Responses for each question were rated on a five-point Likert scale, 1=’not worried at all, 2=’somewhat not worried’, 3=’neutral’, 4=somewhat worried’, and 5=’extremely worried’. The total score of the scale was the sum of the three items, ranging from 3 to 15. A higher total score indicates a greater perceived risk of COVID-19. However, the perceived risk index had Cronbach’s alpha of 0.50, which is very low. The study participants were also asked 12 questions related to their worry about the current COVID-19 crisis, such as losing someone they love due to the disease, health system being overwhelmed, their mental and physical health, and so on., on a three-point scale, where 1=’don’t worry at all’, 2=’worry somehow’, and 3=’worry a lot’. The total score of the scale ranged from 12 to 36. The internal consistency (reliability) of the questions was tested by applying Cronbach’s alpha and the coefficient of the reliability of scale was estimated at 0.91, which is highly acceptable.”

Three more references are added:

15.Brewer NT, Chapman GB, Gibbons FX, Gerrard M, McCaul KD, Weinstein ND. Meta-analysis of the relationship between risk perception and health behavior: the example of vaccination. Health Psychol. 2007 Mar;26(2):136-45. doi: 10.1037/0278-6133.26.2.136. PMID: 17385964.

16.McCarthy-Larzelere, M., Diefenbach, G. J., Diefenbach, D. A., Netemeyer, R. G., Bentz, B. G., & Manguno-Mire, G. M. (2001). Psychometric properties and factor structure of the Worry Domains Questionnaire. Assessment, 8(2), 177–191. https://doi.org/10.1177/107319110100800206.

17.WHO. Regional Office for Europe. Survey tool and guidance: rapid, simple, flexible behavioral insights on COVID-19. 2020. WHO/EURO:2020-696-40431-54222.

https://apps.who.int/iris/handle/10665/333549. 

Comment 4: How you control the bias associated with self-administered questioner?

Authors’ response: Thank you for this comment that refers to the issue of bias with the self-administered questionnaire. The self-administered questionnaire is indeed affected by different biases, particularly by information bias, 

We have tried to deal with this issue of bias in the Methods section, which is stated in the original manuscript as follows (Lines 175 to 191, Page 9):

“A total of 12 experienced data collectors with health backgrounds were involved in the data collection. The research team developed a guideline to guide the data collectors and supervisors for data collection, quality assurance of data and ethical conduct. Training and orientation on the survey tool and methodology, including how to administer the questionnaire was conducted for the data collectors and supervisors using a webinar on 2nd June 2020. After explaining the purpose of the study and obtaining written or oral informed consent, study participants were given a paper-based questionnaire at their workplace and the respondents filled out their questionnaires.” 

“The purpose of the study was clearly stated in the questionnaire, and the study participants were asked to complete the questionnaire with honest answers. They were encouraged to fill out the questionnaire while the data collectors were still in the hospital during the data collection period. A collection center was also prepared in the Hospital Director’s office to gather the questionnaires from the healthcare workers who were unable to deliver the completed questionnaires to the data collectors directly. The data collection took place simultaneously in the six hospitals. The questionnaires were checked for completeness, and consistency upon collection. All responses were anonymous.”

In addition, we have modified our limitation section by adding a sentence that discusses the bias associated with self-administered questionnaire, and we modified the statement as follows: 

“At last, the results of this study might be affected by information bias since it is based on self-reported data using self-administered questionnaire, and the respondents might overestimate or underestimate the responses in a way that they believe is socially acceptable rather than reporting actual or genuine answers.” (Lines 492 to 497, Page 28). 

Comment 5: What were the data quality measures you considered in the study?

Authors’ response: Thank you for this comment that refers to the measures taken to improve the data quality. From the inception of the study, we had concerns about the data quality and bias issues due to our inability to conduct face-to-face interviews due to COVID-19 social distancing rules, and as a result we opted to apply the self-administered data collection approach, which was the only feasible option at the time of the study. The online data collection option was also not feasible since most of the study participants might not have access to the Internet with good connectivity and the study only targeted healthcare professionals in the six hospitals. 

As a result, we finally decided to use a paper-based self-administered data collection method, and we developed a guideline that was used by the data collectors and supervisors in the recruitment of the respondents, handing out, and recollecting the questionnaires. The main components of the guideline that addressed the data quality measures to be undertaken by the supervisors and data collectors are summarized below:

• “Obtain the number of all healthcare workers practicing in the hospital as well as the number working in the different wards. 

• Identify potential respondents from the selected hospital. 

• Introduce your-self to the respondent and provide adequate information about the objectives of the study and data collection procedures. 

• Provide adequate information to the respondent about the ethical issues of this study. 

• Request the willingness/voluntariness of the respondent to get consent to take part in this study. 

• Adequately respond to the questions that might be raised by the respondent. 

• Provide the questionnaire to the respondent with adequate information about the nature of the questions and procedures for completing the questionnaire with honest answers. 

• Get contact information about the respondent such as phone number to remind him/her in case the respondent forgets or lost the questionnaire using a logbook. 

• Provide clear instructions for potential respondents for returning the completed questionnaire to the data collector or other point of collection. 

• Get appointment from the respondent when and where to collect the completed questionnaire using a logbook. 

• Collect the completed questionnaires as per the scheduled appointment. 

• Check all the returned questionnaires for completeness before leaving the respondent to ask or clarify some questions or problems. 

• Develop a strategy for addressing non-response, including how many attempts to reach the respondents face-to-face, by phone and/or other possible methods. 

• List the number of wards, the number of questionnaires distributed and the number of completed questionnaires collected using MS Excel Sheet. 

• At the end of the data collection, prepare and submit 3-4 pages field report about the data collection situation including challenges encountered and solutions made. 

• Submit the completed questionnaires on time along the field report and MS Excel containing the lists of the works done.” 

In addition, 

• The questionnaire was developed and administered in English. There was no need of translation into the local language and back to English as the respondents can easily understand and fill in the questionnaire in English.

• The questionnaire was handed out to the potential respondents with a clear cover letter that state the objective of the survey and highlights why and how they were selected in addition to the information orally provided by the data collectors. 

• Data were checked for completeness, cleaned and entered into the CSPro software by experienced data entry clerks under the supervision of a statistician.

• In this study, higher response rate (92%) was obtained which might reduce the risk of selection bias and enhance validity. Of 1228 potential participants received the questionnaires, 1134 completed the questionnaires. 

Comment 6: The text has a lot of typo and grammatical mistakes. Authors should thoroughly check and improve the language of the manuscript. Some of the text needs to be revised to tone down the narrative. The errors are too numerous to be listed here.

Authors’ response: Thank you for the comments, and we are very sorry about that. The main reason for this was the urgency of the study. We now have carefully reviewed each grammar and editorial issues line by line and made several editorial works and corrections. We have also revised many of the statements and texts in the revised manuscript.

Comment 7: Why you use logistic regression? I don’t agree with you about your analysis method because for this study liner regression is appropriate, why not you use this?

Author’s response: Thank you for the comments. We used logistic regression to make simple the interpretation of the coefficients, as odds ratio, and at the same time some of the assumptions of the linear regression model. We have now used linear regression method, since the effect of minor violation of the assumption on the result will be minimal as the sample size is large.

Comment 8: Please synthesize the result section specifically in characteristics of the study participants. The statistical analysis section (Table-6): Look at the value of the COR and AOR almost all the AOR>COR how this happened and do you trust this value. For me both sampling and systematic error are introduced in your result please see it once again. I have a serious doubt.

Author’s response: Thank you for the comments. Actually, not almost all the AOR>COR, rather than the sampling and systematic sampling, the loss of information by dichotomizing the continuous measurement may have some effect. We have now used linear model to address the above two comments. 

Comment 9: What is the policy implication of this study and what each finding it implies, please enlist in the discussion section.

Author’s response: Thank you for these comments. We have revised the discussion section and highlighted some important implications of the study, for example, as one of the paragraphs indicate below:

“Overall, the participants of the present study indicated higher perceived risk and worry about the COVID-19 crises, which could ultimately affect motivation and performance related to their clinical practice, particularly treating COVID-19 patients. Although the prevalence of preventive measures among the respondents was very high, active interventions such as the provision of adequate PPE and psychological support for HCPs should be considered not only for frontline healthcare workers, but also for all categories of health workers”. (Lines 480 to 485, Page 27)

Comment 10: Concluding remarks are also not very convincing and require more careful evaluation by authors to reflect the message of this paper.

Authors’ response: Thank you for the comment. We revised the Conclusion of the main text as follows:

“In conclusion, our study has illuminated the widespread practices of preventive measures, higher levels of perceived risk and worry about the COVID-19 crises among HCPs who have direct or indirect contact with COVID-19 patients. The study showed a universally higher preventive practices among the study participants to prevent SARS-CoV-2 infections. The higher level of worry about COVID-19 might be helpful in improving the perceived risk of the pandemic among the HCPs, which can motivate them to adopt proper preventive measures. This can be addressed through the implementation of risk communication programs with the public and healthcare workers during the current COVID-19 pandemic.” (Lines 500 to 507, Page 28)

Comment 11: Implications for practice, research, and theory were NOT clearly discussed. –Recommendations are targeted to the stakeholders and founded on the findings gained from the study.

Authors’ response: Thank you for the comment. We revised the last two sentences of the Conclusion of the main text as follows:

“The higher level of worry about COVID-19 might be helpful in improving the perceived risk of the pandemic among HCPs, which can motivate them to adopt proper preventive measures. This can be addressed through the implementation of risk communication programs with the public and healthcare workers during the current COVID-19 pandemic.” (Lines 500 to 507, Page 28)

 6. PLOS authors have the option to publish the peer review history of their article (what does this mean?). If published, this will include your full peer review and any attached files.

Do you want your identity to be public for this peer review? For information about this choice, including consent withdrawal, please see our Privacy Policy.

Reviewer #1: Yes: Dr. Muhammad Abaid Ullah

Reviewer #2: No

---

## [Editor Report · Decision Letter 1]

16 Jun 2021

Risk perceptions and preventive practices of COVID-19 among healthcare professionals in public hospitals in Addis Ababa, Ethiopia

PONE-D-20-33821R1

Dear Author,

We’re pleased to inform you that your manuscript has been judged scientifically suitable for publication and will be formally accepted for publication once it meets all outstanding technical requirements.

Kind regards,

Ramesh Kumar, PhD

Academic Editor

PLOS ONE
---

## [Editor Report · Acceptance letter]

18 Jun 2021

PONE-D-20-33821R1 

Risk perceptions and preventive practices of COVID-19 among healthcare professionals in public hospitals in Addis Ababa, Ethiopia 

Dear Dr. Deressa:

I'm pleased to inform you that your manuscript has been deemed suitable for publication in PLOS ONE. Congratulations! Your manuscript is now with our production department. 

Kind regards, 

on behalf of

Dr. Ramesh Kumar 

Academic Editor

PLOS ONE